# Pain scores reduction with the use of ultrasound-guided paracervical nerve block in patients with cervical cancer undergoing intracavitary brachytherapy: A randomized controlled trial

**Yuanyuan Rong[1], Yi Yang[2], Xi Zhang[1], Xiaoxiao Zhou[2], Jianfeng Fu[1], Xuelian Zhao[1], Huaqin Liu[1], Huanshuang Pei[1], Chao Zhou[1]***

1 Department of Anesthesiology, The Fourth Hospital of Hebei Medical University, Shijiazhuang, Hebei, China, 2 Department of Gynecology and Obstetrics Ultrasound, The Fourth Hospital of Hebei Medical University, Shijiazhuang, Hebei, China

* zhouchao870607@163.com

## Abstract

### Study objective

To determine the safety and effectiveness of ultrasound-guided paracervical nerve blocks for the painless treatment of patients with cervical cancer post-implantation.

### Design

Single-center randomized controlled trial.

### Setting

Fourth Hospital of Hebei Medical University (July 2023 to October 2023).

### Trial number

ChiCTR2300071580 [https://clin.larvol.com/trial-detail/ChiCTR2300071580].

### Patients

Eighty patients with cervical cancer underwent post-implantation treatment.

### Interventions

Patients receiving and not receiving paracervical nerve blocks (Groups T and C, respectively) were randomly allocated.

### Measurements

Primary measures included visual analog scale (VAS) scores and patient body movement scores at various stages, including vaginal speculum placement (T1), applicator/needle

**Data Availability Statement:** All relevant data are within the paper and Supporting information files.

**Funding:** This study was supported by the Key Scientific and Technological Research Plan of the Hebei Health and Health Commission [grant number 20241497]. Yuanyuan Rong received funding. The website is http://wsjkw.hebei.gov.cn/ The funder dose not play any role in the study design, data collection and analysis, decision to publish, or preparation of the manuscript.

**Competing interests:** The authors have declared that no competing interests exist.

insertion (T2), treatment administration following connection of the treatment tube (T3), needle withdrawal and hemostasis (T4), and willingness to receive further treatment. The secondary observation indices in this study included the operation time, incidence of hypoxemia, occurrence of nausea and vomiting, adverse events related to the circulatory system, patient satisfaction score, operator satisfaction score, and operation duration required by patients with an Alderte score of $\geq 9$.

## Main results

Forty patients each were randomly allocated into Groups T and C. The VAS scores did not differ significantly between the two groups at T1. However, at T3 and T4, the VAS scores of Group T was significantly lower than that of Group C. No significant difference was observed in the body movement scores between the two groups at T1 and T3. However, the body movement score of Group T was significantly lower than that of Group C ($P < 0.001$) at T2 and T4. Group T showed higher postoperative satisfaction and willingness to receive further treatment compared to that of Group C.

## Conclusions

Ultrasound-guided paracervical nerve block effectively reduced the pain scores in patients with cervical cancer undergoing post-implantation treatment and enhanced their inclination to undergo further treatment.

## 1. Introduction

Cervical cancer has the highest mortality and morbidity rates among all cancers worldwide [1]. Patients with cervical cancer benefit greatly from brachytherapy, which is an important component of radical radiotherapy [2, 3]. Intracavitary brachytherapy (ICBT), interstitial brachytherapy (ISBT), and hybrid intracavitary/interstitial brachytherapy are the three main types of brachytherapy [4]. ISBT can be used in combination with ICBT to compensate for the insufficient coverage of tumor tissues irradiated by ICBT at high doses. ISBT in combination with ICBT enables the treatment of lesions in patients in whom ICBT alone cannot provide adequate improvement [5]. Moreover, ISBT requires the insertion of an implantation needle into the cervical tumor, which can cause considerable pain [6]. Patients with cervical cancer reportedly experience nausea, vomiting, and other symptoms of discomfort during needle insertion into the cervix [7, 8]. In severe cases, patients may experience bradycardia; hypotension; and increased muscle tension in the lower limbs, hip oscillations, and reflexive leg movements due to hyperactivity of the vagus nerve [6]. Additionally, patient movement may result in insufficient tumor irradiation, reduced local control and survival rates, and increased incidence of complications [9]. Pain may also affect the patients' quality of life resulting in chronic pain, anxiety, and depression, as well as discourage them from undergoing further brachytherapy [10–13]. Patients who receive ISBT are shifted between rooms for simulated positioning, treatment, and surgery. Thus, anesthesiologists find it challenging to relieve patient pain during these transfers. Therefore, it is imperative to elucidate methods for reducing pain in patients during ISBT and enhancing their willingness to receive further treatment.

Previous research has demonstrated that the administration of a parasympathetic nerve block can effectively impede the transmission of pain signals originating from the uterine body

and the upper regions of the cervix and vagina [14], thereby mitigating or eradicating the pain experienced by patients. Nevertheless, studies examining the efficacy of this technique for pain management following cervical cancer implantation are lacking. Thus, we hypothesized that ultrasound-guided paracervical nerve blocks may reduce the pain scores of patients undergoing ISBT and increase their willingness to receive further treatment. We then conducted a randomized controlled trial to validate the effectiveness and safety of ultrasound-guided paracervical nerve blocks in patients with cervical cancer undergoing ISBT.

## 2. Materials and methods

### 2.1. Ethics

This single-center randomized controlled trial was approved by the Ethics Committee of the Fourth Hospital of Hebei Medical University (Approval number: 2022154). The research protocol was registered with the China Clinical Trial Network Registration Center (registration number: ChiCTR2300071580) on May 18, 2023. Written informed consent was obtained from all patients before the trial. This manuscript adheres to the Consolidated Standards of Reporting Trials (CONSORT) guidelines [15]. The CONSORT flow diagram of this study is as follows (Fig 1). The authors declare that all experiments on human subjects were conducted in accordance with the Declaration of Helsinki and that all procedures were carried out with the adequate understanding and written consent of the subjects.

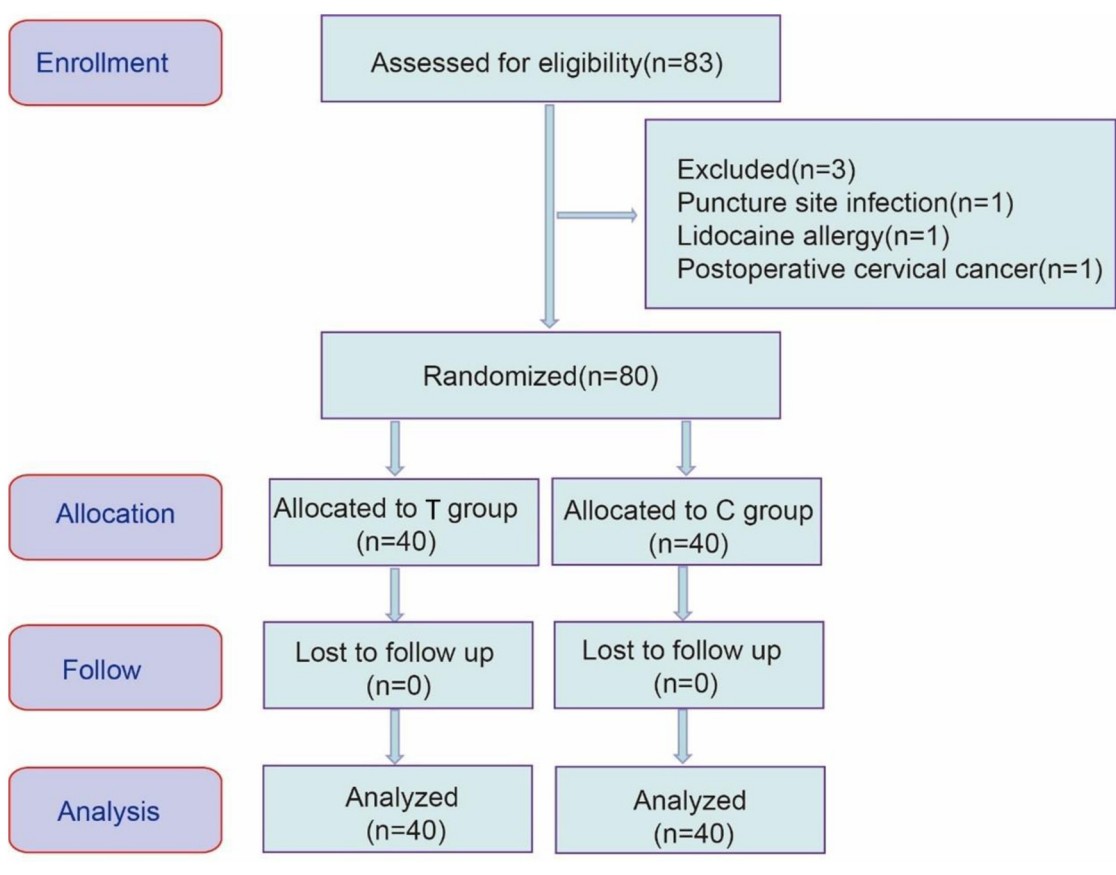

**Fig 1. The CONSORT flow diagram.**

## 2.2. Participants

We recruited patients who received non-invasive post-implantation treatment for cervical cancer and anticipated interstitial implantation radiotherapy at the Fourth Hospital of Hebei Medical University between July and October, 2023. Patients were excluded if they showed signs of infection at the puncture site, exhibited an allergic reaction to the experimental drugs, were diagnosed with mental illness, or were unable to cooperate during the surgical procedure. Further exclusion criteria included severe liver and kidney failure or cardiac failure classified as Grade III or above according to the New York Cardiology Association guidelines [16]. Independent researchers executed a balanced random assignment, categorizing eligible patients into either Group T, those receiving the paracervical nerve block, or Group C, the control group containing patients that did not receive the paracervical nerve block. The principal investigator was notified of the group assignments via email prior to the patient's entry into the operating room. Only the anesthesiologist and gynecologist on duty were informed of the group allocation, while the patients and the acute pain service (APS) team responsible for assessing the postoperative results remained blind to the group assignments.

## 2.3. Anesthesia

All the patients underwent preanesthetic evaluation and were required to fast for 6h. Routine monitoring, including electrocardiography, percutaneous pulse oxygen saturation, noninvasive arterial blood pressure evaluation, and bispectral index (BIS) monitoring, were performed in the operating room. A peripheral venous fluid circuit was created and filled with normal saline solution. Thereafter, the patient assumed the lithotomy position, and the gynecologist sterilized the sheets, positioned the vaginal speculum, and initiated intravenous administration of 0.2–0.3 mg/kg remimazolam and 0.03 mg/kg nalbuphine under general anesthesia, while maintaining a BIS value of 40–60. During implantation, each patient received additional doses of 0.05 mg/kg remimazolam, calibrated in response to the body movement and BIS depth to maintain sedation. The minimum interval between two doses was 1 min. If the anesthesia depth was found to be insufficient, additional intravenous injections of 0.5 mg/kg propofol were administered until the BIS value reduced to ≤ 60. Each patient was administered oxygen through a nasal catheter at a concentration of 100% and flow rate of 4 L/min for the entire duration of treatment. Bilateral paracervical nerve blocks were performed in Group T using B-ultrasound with 2% lidocaine (2.5 ml per side), whereas no nerve blocks were performed in Group C. Anesthesiologists administered the paracervical nerve blocks, after which gynecologists performed the tissue implantation. Flumazenil (0.5 mg) was administered after positioning the vaginal gauze pack. Upon achieving an Alderte score ≥ 9, computed tomography was performed for implant needle positioning and evaluation, following which the patients were moved to the treatment room for afterloading radiotherapy. After radiation, the needle was extracted, and hemostasis was achieved through compression; the patient was discharged after ensuring the absence of any complications.

The paracervical nerve block method involved the use of a 3–10 MHz electronic convex array intracavity probe, which was positioned horizontally at the 3 o'clock position within the paracervical vault for visualization of the uterine artery located in the deeper region of the vault as the paracervical nerve surrounds the uterine artery. To administer the block, an ultrasound-guided in-plane technique was employed using an 18G×200 mm puncture needle. The needle was inserted at a targeted depth of approximately 1.5–2 cm into the uterus and 2.5 ml of 2% lidocaine was injected. The same procedure was performed after placing the probe horizontally at the 9 o'clock position in the paracervical vault (Fig 2).

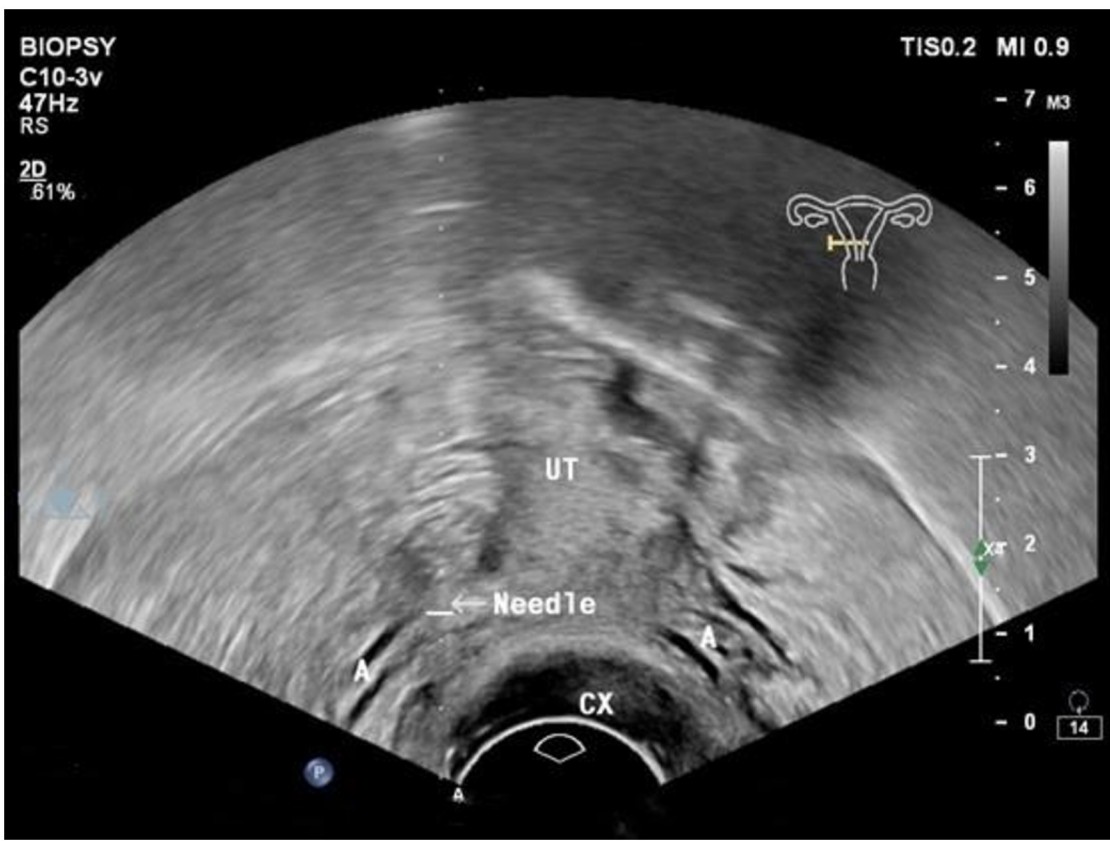

**Fig 2. Illustration of the procedure of ultrasound-guided paracervical nerve block.** A, Uterine artery; UT, uterus; CX, cervix.

Upon arrival in the operating room, vital statistics, including heart rate, blood pressure, and pulse oxygen saturation, were recorded for each patient. Blood pressure was monitored every 3 min throughout the examination process, with a decrease in the average arterial pressure by 30% of the baseline values indicating hypotension prompting the administration of 3 mg ephedrine, which was noted in the medical records. Sinus bradycardia, which is defined as a heart rate < 50 beats/min, was treated with atropine (0.5 mg) and supplemented with ephedrine (3 mg) in cases of simultaneous hypotension. The visual analog scale (VAS) scores and patient body movement scores were recorded at the following stages of post-implantation: vaginal speculum placement, T1; applicator/implantation needle, T2; treatment administration following connection of the treatment tube, T3; and needle withdrawal and hemostasis, T4.

The principal outcome measures were VAS scores and patient body movement scores at various stages (T1, T2, T3, and T4). The secondary observations included operation duration; occurrence of hypoxemia, nausea, and vomiting; and circulatory system related adverse events, such as abnormal heart rate (< 50 or > 120 beats/min) and mean arterial pressure (MAP) exceeding 30% of the preoperative measures. Additionally, patient satisfaction score, operator satisfaction score, and duration of surgery in patients with an Aldrete score > 9 min were also assessed. Members of the APS team conducted periodic visits at various operative stages to estimate the VAS scores. In addition, the anesthesiologist in charge monitored the patient's body movements during the operation. The aggregate VAS scores translates to 10 points, where 0 indicates no pain; 1–3, slight pain tolerable by the patient; 4–6, tolerable pain affecting

sleep; and 7–10, intense, unbearable pain. The patients' body movements were scored as 1, no physical activity; 2, slight physical activity not hindering operational proceedings; or 3, major body motion disrupting the operator's workflow. Both the patients' and operators' satisfaction were scored on a 4-points scale; 1 being "very satisfied" and 4 being "very dissatisfied" for patients, and 1 as "very satisfactory" to 4 as "highly unsatisfactory" for operators.

## 2.4. Statistical analysis

During the pre-trial phase, VAS scores $\geq 3$ were observed in 40% and 12% of the cases in Group C and Group T, respectively. Upon employing the comparison formula for two independent sample rates, $n = 0.5 \times \left[ \frac{(u_\alpha + u_\beta)}{sin^{-1}\sqrt{p_1} - sin^{-1}\sqrt{p_2}} \right]^2$, it was inferred that each group required the inclusion of 36 participants (with a power of 0.8 and a type I error of 0.05, $u_\alpha$ and $u_\beta$ correspond to *the $\mu$* value of type I and type II errors, respectively, $u_\alpha = 1.96$; $u_\beta = 0.842$). To mitigate potential issues, such as rejection and attrition, a total sample size of 40 patients per group was considered.

The Mann–Whitney U test was used to compare the VAS score at three different time points and the patient's body movement score at four different time points between the two groups. A t-test was used to compare the normally distributed data. Additionally, the Mann–Whitney U test was used to compare the continuous variables. The parametric results are expressed as the mean standard deviation, while the nonparametric results are expressed as medians [interquartile interval]. All the reported *P*-values were two-tailed, and those < 0.05 were considered statistically significant. Statistical analyses were conducted using SPSS Statistics 23.0 (IBM Corp., Armonk, NY, USA).

## 3. Results

A total of 80 eligible patients were randomly allocated between July and October, 2023 into two groups based on the predetermined inclusion and exclusion criteria. All the patients successfully underwent the ISBT procedure without experiencing any severe complications, such as ureteral injury, perforation, or bleeding. Notably, no statistically significant differences were observed in the age, body mass index, American Society of Anesthesiologists grading ratio, heart rate, and MAP at different time points between the two groups (*P* > 0.05) (Tables 1–3).

No statistically significant difference was observed in the VAS score at T1 between the two groups (*P* > 0.05). However, the VAS scores of Group T were significantly lower than those of Group C (*P* < 0.001) at T3 and T4 (Table 4). No significant difference was observed in the body movement scores between the two groups at T1 and T3 (*P* > 0.05). Nevertheless, the body movement scores of Group T were significantly lower than those of Group C (*P* < 0.001) at T2 and T4 (Table 5).

No significant differences were observed in remimazolam, additional remimazolam, and nalbuphine dosage between the two groups (*P* > 0.05). However, the number of propofol

**Table 1. Comparisons of the general conditions of the two groups of patients (n = 40).**

| Groups | Age (years, mean ± SD) | BMI (kg/m², mean ± SD) | ASA I/II/III (cases) |
|---|---|---|---|
| Group T | 50.28±9.90 | 23.28±3.61 | 5/33/2 |
| Group C | 51.88±11.31 | 23.95±4.14 | 7/32/1 |

ASA, American Society of Anesthesiologists grading ratio; BMI, body mass index; Group C, control group; Group T, test group; SD, standard deviation

**Table 2. Comparison of heart rate between groups at different time points (n = 40, mean ± SD).**

| Groups | T1 | T2 | T3 | T4 |
|---|---|---|---|---|
| Group T | 89.2±8.89 | 83.85±11.00 | 82.05±11.35 | 84.93±10.76 |
| Group C | 91.18±14.92 | 84.93±12.02 | 85.43±11.93 | 87.70±11.93 |
| Difference T-C (95% CI) | -1.98 (-7.44~3.49) | -1.08 (-6.20~4.05) | -3.38 (-8.58~1.81) | -2.78 (-7.83~2.28) |
| *P*-value | 0.47 | 0.68 | 0.20 | 0.28 |

Data for Group T and C are presented as mean ± SD

CI, confidence interval; Group C, control group; Group T, test group; SD, standard deviation; T1, vaginal speculum placement; T2, applicator/needle insertion; T3, treatment administration following connection of the treatment tube; T4, needle withdrawal and hemostasis

**Table 3. Comparison of MAP between groups at different time points (n = 40, mean ± SD).**

| Groups | T1 | T2 | T3 | T4 |
|---|---|---|---|---|
| Group T | 91.88±14.37 | 84.33±17.64 | 87.43±12.30 | 90.78±14.49 |
| Group C | 91.63±14.31 | 83.50±13.63 | 84.80±11.95 | 88.38±11.73 |
| Difference T-C (95% CI) | 0.25 (-6.13~6.63) | 0.83 (-6.19~7.84) | 2.63 (-2.77~8.02( | 2.4 (-.347~8.27) |
| *P*-value | 0.94 | 0.82 | 0.34 | 0.42 |

Data for Group T and C are presented as mean ± SD

CI, confidence interval; Group C, control group; Group T, test group; SD, standard deviation; T1, vaginal speculum placement; T2, applicator/needle insertion; T3, treatment administration following connection of the treatment tube; T4, needle withdrawal and hemostasis

**Table 4. Comparison of VAS between groups at different time points (n = 40).**

| Groups | T1 | T2 | T3 | T4 |
|---|---|---|---|---|
| Group T | 0 (0,1) | / | 0 (0,1) | 1 (0,2) |
| Group C | 0 (0,2) | / | 1 (0,2) | 4 (3,5) |
| *P*-value | 0.48 | / | <0.001 | <0.001 |

Data for Group T and C are presented as median (Q1,Q3)

Group C, control group; Group T, test group; Q, quartile; T1, vaginal speculum placement; T2, applicator/needle insertion; T3, treatment administration following connection of the treatment tube; T4, needle withdrawal and hemostasis

**Table 5. Comparison of body movement scores between groups at different time points (n = 40).**

| Groups | T1 | T2 | T3 | T4 |
|---|---|---|---|---|
| Group T | 1 (1,1) | 1 (1,1) | 1 (1,1) | 1 (1,1) |
| Group C | 1 (1,2) | 2 (1,2) | 1 (1,1) | 2 (1,2) |
| *P*-value | 0.17 | <0.001 | 0.32 | <0.001 |

Data are presented as median (Q1,Q3)

Group C, control group; Group T, test group; Q, quartile; T1, vaginal speculum placement; T2, applicator/needle insertion; T3, treatment administration following connection of the treatment tube; T4, needle withdrawal and hemostasis

**Table 6. Dosages of remimazolam, including supplementary doses, and nalbuphine, along with the supplementary dosage of propofol were assessed in both groups (n = 40).**

| Groups | Remimazolam, mg | Remimazolam supplementary, mg | Nalbuphine, mg | Propofol, mg |
|---|---|---|---|---|
| Group T | 15.5 (13,21) | 0 (0,0) | 2 (1.5,2) | 0 (0,20) |
| Group C | 15 (13.25,18) | 0 (0,0) | 2 (2,2) | 62.5 (20,100) |
| P-value | 0.65 | 0.78 | 0.38 | <0.001 |

Comparison was made between cases with or without propofol administration

Data are presented as median (Q1,Q3)

Group C, control group; Group T, test group; Q, quartile; T1, vaginal speculum placement; T2, applicator/needle insertion; T3, treatment administration following connection of the treatment tube; T4, needle withdrawal and hemostasis

**Table 7. Comparisons between groups concerning operating times, time taken to achieve an Aldrete score ≥9, operator satisfaction, patient satisfaction, and patient willingness to receive further treatment.**

|  | Group T (n = 40) | Group C (n = 40) | Difference T-C (95% CI) | P-value |
|---|---|---|---|---|
| Operating times (min) | 5 [4,6] | 5.5 [5,7] |  | 0.50 |
| Time taken to achieve an Aldrete score ≥9(min) | 2.03±1.19 | 3.35±1.51 | 1.33(0.72~1.9) | <0.001 |
| Operator satisfaction (n) |  |  |  | 0.50 |
| 1: very satisfied | 38 | 39 |  |  |
| 2: satisfied | 2 | 1 |  |  |
| 3: dissatisfied | 0 | 0 |  |  |
| 4: very dissatisfied | 0 | 0 |  |  |
| Patient satisfaction (n) |  |  |  | 0.03 |
| 1: very satisfied | 38 | 30 |  |  |
| 2: satisfied | 2 | 6 |  |  |
| 3: dissatisfied | 0 | 4 |  |  |
| 4: very dissatisfied | 0 | 0 |  |  |
| Willingness to receive further treatment (n[%]) | 40 (100%) | 34 (85%) | 15%(3.18%~29.07%) | 0.03 |

Operating times is presented as median (Q1,Q3)

Time taken to achieve an Aldrete score ≥9 is presented as mean±SD

CI, confidence interval; Group C, control group; Group T, test group

supplements was significantly lower and the dosage was smaller ($P < 0.001$) in Group T compared to Group C (Table 6). Additionally, no significant differences were observed in the operation time and operator satisfaction scores between the two groups ($P > 0.05$). Patients with an Aldrete score ≥ 9 exhibited a shorter duration of treatment (2.03±1.187 vs. 3.35±1.511) in Group T compared to those in Group C. Additionally, patients in Group T demonstrated higher levels of patient satisfaction ($P = 0.031$) and greater willingness to receive further treatment ($P = 0.026$) compared to those in Group C (Table 7).

## 4. Discussion

ISBT is an invasive procedure causing significant pain and other symptoms, such as nausea, vomiting, and discomfort. In severe cases, hyperactivity of the vagus nerve may induce bradycardia and hypotension, which can be accompanied by increased muscle tension in the lower limbs, hip oscillations, and reflexive leg movements; it may even elevate the likelihood of bleeding and perforation [7, 8]. The present study used ultrasound-guided paracervical nerve blocks, targeting nerves from the uterine and pelvic floor plexuses located in the lower uterus

and upper vagina, which would have hindered the transmission of the following two types of pain signals [17]: 1) those emerging from the stimulation of the cervix and upper vagina and excitation of the parasympathetic afferent fibers by the use of instruments, such as the vaginal speculum and applicator, resulting in dull lower back pain via S2–S4 pelvic visceral nerves; 2) those emerging from tissue damage and inflammatory responses in the perineum attributed to the insertion of needles, templates, and custom-made applicators into the peripartum tissue, stimulating the afferent fibers of the somatic nerves and causing stinging sensations via S2–S4 pelvic visceral nerves. Consequently, the VAS scores exhibited by the patients in Group T were lower than those in Group C at both T3 and T4. Previous research has demonstrated that the administration of a local anesthetic into the paracervical plexus through a paracervical block can effectively inhibit nerve conduction, thereby fulfilling the fundamental criteria for painless surgical procedures [18]; it also confirms our viewpoint.

Our research showed that during T3, when the patient was conscious and no surgical procedure was being performed, the elevated VAS scores in Group C was tolerable, resulting in no discernible disparity in the body movement scores between the two groups. In contrast, during T2 and T4, owing to the strong stimulation of the operator's interventions (implantation, extraction, and compression of the implant needle to stop bleeding) on the patients, those in Group C had involuntary body movements under sedation, with higher body movement scores.

Remimazolam, a novel water-soluble benzodiazepine with ultrashort-acting properties, exhibits rapid onset and metabolism [19]. Conscious sedation anesthesia can be used in patients with gynecological tumors undergoing ISBT treatment; however, when the sedation effect disappears and the instrument is removed, patients often exhibit moderate pain, subsequently requiring high doses of analgesics and potential psychosocial treatment [20]. Moreover, following applicator placement, transfer between the operating, imaging, and radiation therapy rooms is necessary. Such repeated movement can further increase the patient's experience of pain stimulation. Our research demonstrates that ultrasound-guided paraventricular nerve blocks can reduce VAS scores and discomfort in patients during treatment, resulting in higher patient satisfaction and willingness to receive further treatment in Group T. Therefore, adopting the anesthesia method of remimazolam sedation combined with ultrasound-guided paracervical nerve block represents a more suitable choice. Consistent with our findings, other studies demonstrated that patients under moderate and deep sedation receiving paracervical block experienced no or mild pain [21].

Propofol inhibits the respiratory and circulatory system; therefore, the use of propofol as a stand-alone agent is associated with a high incidence of hypoxemia and hypotension [22]. Thus, in the present study, propofol alone was considered a remedial intervention rather than a primary treatment. Additionally, the implementation of the ultrasound-guided paracervical nerve block enhanced the precision of both localization and anesthetic efficacy, consequently lowering the occurrence of nerve block-associated complications; hence, no anesthesia-related complications were observed in the present study.

Our research findings have important practical implications, especially given the growing patient populations. With medical centers worldwide facing mounting pressures to uphold the standard of care amidst resource limitations, our findings offer valuable insights into optimizing medical service delivery. In many hospitals, shortages of anesthesia equipment and professional anesthesiologists, and challenging economic circumstances hinder the implementation of optimal anesthesia strategies. Hospitals have expressed the need for safe, convenient, effective, and inexpensive anesthetic techniques. Notably, our proposed anesthesia method effectively mitigates patient pain and achieves surgical requirements to a considerable extent while simultaneously meeting the above qualities.

Despite these strengths, our study has several limitations. First, as the study was conducted within a single center with a comparatively small sample size, the conclusions drawn remain limited only to the reduction in the pain score of patients undergoing ISBT and enhancing their willingness to receive further treatment; therefore, the potential impact of additional brachytherapy procedures warrants extensive investigation through larger multicenter studies. Second, while we did not encounter complications associated with the use of the paracervical nerve block in our study owing to the small sample size, this does not unequivocally confirm their absence; a large multicenter study needs to be conducted to ascertain the safety profile. Third, several patients in our study experienced discomfort despite receiving paracervical nerve block, which could potentially be associated with the short duration of action of lidocaine; thus, substitution with long-acting local anesthetics warrants further investigation. Fourth, this study did not account for qualitative factors, such as anxiety or fear, associated with the patient's pain experience. These sentiments can significantly affect the patient's membrane-sweeping experience, influence compliance, and induce psychological distress [23].

Henceforth, we will conduct more in-depth research to consolidate the present findings. Firstly, we will conduct larger sample sizes and multi-center trials to verify the reliability and universality of these data. Secondly, we consider that studying the long-term effects of paracervical nerve block on patient prognosis and quality of life would be valuable. In addition, evaluating the cost-effectiveness and potential side effects of this technique will be another focus of future research. Finally, we hope to combine various techniques including nerve block to provide better pain relief for patients with cervical cancer.

In conclusion, ultrasound-guided paracervical nerve blocks reduced the pain scores in patients with cervical cancer following post-implantation treatment, thereby increasing their willingness to receive further treatment.

## Supporting information

**S1 Checklist. CONSORT 2010 checklist of information to include when reporting a randomised trial\*.**
(DOC)

**S1 Data.**
(XLSX)

## Author Contributions

**Conceptualization:** Yuanyuan Rong, Huaqin Liu, Huanshuang Pei, Chao Zhou.

**Data curation:** Yi Yang, Xi Zhang, Xiaoxiao Zhou, Chao Zhou.

**Methodology:** Jianfeng Fu, Xuelian Zhao.

**Project administration:** Jianfeng Fu, Xuelian Zhao.

**Software:** Chao Zhou.

**Supervision:** Huaqin Liu, Huanshuang Pei.

**Validation:** Huaqin Liu, Huanshuang Pei.

**Visualization:** Yi Yang, Xiaoxiao Zhou.

**Writing – original draft:** Yuanyuan Rong.

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
