## [Decision Letter · Decision Letter 0]

15 Mar 2024

PONE-D-23-40504Pain Score reduction with the use of ultrasound-guided paracervical nerve block in patients with cervical cancer undergoing intracavitary brachytherapy: A randomized controlled trialPLOS ONE

Dear Dr. Zhou,

Thank you for submitting your manuscript to PLOS ONE. After careful consideration, we feel that it has merit but does not fully meet PLOS ONE’s publication criteria as it currently stands. Therefore, we invite you to submit a revised version of the manuscript that addresses the points raised during the review process.

The manuscript has been evaluated by two reviewers, and their comments are available below. The reviewers have raised a number of major concerns. Reviewer #1 has several recommendations to improve the quality of the statistical analysis, whilst Reviewer #2 raises questions about the clinical appropriateness of this intervention, which raises questions about both the academic contribution of this study and the validity of the conclusions; therefore, please ensure you provide detailed clinical justifications for this intervention and ensure the conclusions are presented appropriately. Please also provide additional details about your inclusion criteria in the Methods.

Could you please carefully revise the manuscript to address all comments raised?

We look forward to receiving your revised manuscript.

Kind regards,

Marianne Clemence

Staff Editor

PLOS ONE

Journal Requirements:

2. Thank you for your submission to PLOS ONE. We note that the protocol uploaded with your submission is written in the past tense and includes exact study dates. Could you please confirm that the clinical trial protocol you have included as Supplementary Information is the version that was submitted to and approved by your ethics committee before the trial began? By the clinical trial protocol, we mean the complete and detailed plan for the conduct and analysis of the trial that was approved by the ethics committee. Note that we do not accept summaries of the clinical trial protocol or published reports of the protocol. Please provide the protocol in the original language. If it is in a language other than English, please also provide a translation. (We will accept a translation of the main points of the protocol.) Your study protocol will be made available to the editors and reviewers, and will be published as supporting information with your manuscript if accepted for publication.

Reviewers' comments:

Reviewer's Responses to Questions

**Comments to the Author**

1. Is the manuscript technically sound, and do the data support the conclusions?

Reviewer #1: Yes

Reviewer #2: No

2. Has the statistical analysis been performed appropriately and rigorously? 

Reviewer #1: No

Reviewer #2: Yes

3. Have the authors made all data underlying the findings in their manuscript fully available?

Reviewer #1: Yes

Reviewer #2: Yes

4. Is the manuscript presented in an intelligible fashion and written in standard English?

Reviewer #1: Yes

Reviewer #2: Yes

5. Review Comments to the Author

Reviewer #1: Major Revision Required

This trial seems a very worthwhile, well designed and conducted trial. However, the analysis needs to be re-examined.

Although, there is considerable interest in the differences between randomised groups at T1, T2, T3 and T4 the following overall comparison may be useful to include.

Assuming T1 to T4 are (at least approximately) at equal time intervals apart, then an overall mean VAS value for each patient could be calculated. The means of these for each group can then be obtained as (say) VL and VC. Following this the difference d = VC-VL can be obtained and a 95% Confidence Interval (CI) can be calculated.

If the T1 to T4 cannot be assumed equally space then the regression slope for each patient can be calculated and these used as the individual data points to be compared in the above. However, this is unlikely to be necessary and often gives the same conclusions as the previous paragraph approach.

Some other points are listed below:

Page Lines

2 24 Not sure why the labels L and C are chosen for the Test and Control treatments – although the reason for C as control is obvious (see comments on Table 1 below)

11 190 As typed the expression for the sample size for one arm of the trial is not correct. The 164.4 should be replaced by 0.5. See for example, Dobson and Gebski (1982), The Statistician, 35, 51-53. However, the sample size calculated of 40 per group (80 in all) is appropriate.

190 Note if the equation is to be quoted, definitions of what uα and uβ represent are required.

12 213 Table 1 Column 1 rows may be better labelled PCN, Test, Control or PCN, Yes, No.

Table 1 As the L and C groups are allocated at random, statistical testing is not required for comparing differences of age, BMI and ASA as, by definition, such differences are random. So, best to omit the P-value row.

Table 1 As this is describing the patients, rather than the SD better to quote the minimum and maximum values of each variable

12 Table 2 As the focus of the trial is on the difference L − C, this table should give these differences for each of T1 to T4, that is, -1.98, -1.08, -3.38 and -2.77. Then quote the 95% CI for each of these differences plus the P-values already quoted.

Usual to quote P-values to 2 significant figures, hence 0.47, 0.68, 0.20 and 0.29. This comment also applies to other places in the text.

12-13 Table 3 Same comments as for Table 2

13 Tables 4 & 5 I think it would be more helpful to quote the minimum and maximum scores rather than Q1 and Q3.

14-15 Tables 6 & 7 More helpful to quote the minimum and maximum scores rather than Q1 and Q3.

15 Table 7, row 2 Time taken by patients with high Alderete scores refers to a subset of the 80 patients, in which case the numbers concerned in each group need to be quoted here.

Also, the difference in groups needs to be quoted with the 95% CI. Since the distribution of time is very skew, comparisons of medians are required. I do not know if these can be implemented in SPSS Statistics 23.0 which is used by the authors for their analysis.

Alternatively, a t-test can be used on the log transformations of the individual patient times.

Reviewer #2: This is a well-written but not thought-provoking study which trying to assess the effect of ultrasound-guided paracervical nerve block in patients with cervical cancer undergoing intracavitary brachytherapy. There are some possible problems of the article which I have to make comment.

1. This type of treatment procedure has a very short duration, and in the past, we have achieved good results by combining short-term sedatives with low-dose analgesics. However, in this study, the two groups did not routinely use analgesic drugs, so it is reasonable for the control group to have a higher VAS score in T4.

2. Compared to administering low-dose analgesics, ultrasound-guided paracervical nerve block requires additional equipment, operator capabilities, and more potential complications. Is it worth doing this in clinical work? Whether nerve block is better than local anesthesia or the use of low-dose analgesics should be more discussed. If a simpler method can be used to achieve the same effect, nerve block is obviously a more complex and risky operation.

3. Is it common to use BIS monitoring for only 5 minutes of operation? Is it necessary or a waste of resources?

6. PLOS authors have the option to publish the peer review history of their article (what does this mean?). If published, this will include your full peer review and any attached files.

Reviewer #1: No

Reviewer #2: No

---

## [Author Response · Author response to Decision Letter 0]

2 Apr 2024

2024/4/2

Marianne Clemence

Staff Editor

PLoS One

Dear Editors:

We wish to re-submit the manuscript entitled “Pain score reduction with the use of ultrasound-guided paracervical nerve block in patients with cervical cancer undergoing intracavitary brachytherapy: A randomized controlled trial.” The manuscript ID is PONE-D-23-40504. 

We thank you and the reviewers for your thoughtful suggestions and insights. The manuscript has benefited from these insightful suggestions. I look forward to working with you and the reviewers to move this manuscript closer to publication in the PLoS One.

We have studied the comments carefully and have made the necessary corrections in accordance with the reviewers’ suggestions. Revised portions of text are marked in red in the manuscript. The responses to all comments have been prepared and given below. 

Thank you for your consideration. I look forward to hearing from you.

Sincerely,

Chao Zhou,

Department of Anesthesiology, The Fourth Hospital of Hebei Medical University

zhouchao870607@163.com

+8618531179002

Response to Journal Requirements:

Q1: Please ensure that your manuscript meets PLOS ONE's style requirements, including those for.

Response: We have checked the manuscript formatting and file naming protocols to ensure compliance with PLOS ONE's style requirements. If any errors remain, please do not hesitate to inform us.

Q2: We note that the protocol uploaded with your submission is written in the past tense and includes exact study dates. Could you please confirm that the clinical trial protocol you have included as Supplementary Information is the version that was submitted to and approved by your ethics committee before the trial began?

Response: In the newly uploaded translated, we have corrected the tense issues. Additionally, we have provided the original version of the research protocol as evidence. We confirm that the clinical trial protocol included as Supplementary Information is the version that underwent submission to to and approval by our ethics committee before the commencement of the trial.

Responses to the reviewer's comments:

Reviewer ＃1: 

Q1: Page 2 Line 24. Not sure why the labels L and C are chosen for the Test and Control treatments – although the reason for C as control is obvious (see comments on Table 1 below). 

Response: Accordingly, we have changed the label of the test group to “T,” corresponding to “Test,” to make it easier for readers to understand.

Q2: Page 11 Line 190. As typed the expression for the sample size for one arm of the trial is not correct. The 1641.4 should be replaced by 0.5. See for example, Dobson and Gebski (1982), The Statistician, 35, 51-53. However, the sample size calculated of 40 per group (80 in all) is appropriate.

Response: Thank you for highlighting this issue. We have conducted a thorough searched for related literature and have not found any errors in this formula. An image of the formula is shown below for reference. The discrepancy between the two formulas, despite producing identical results, stems from the unit used to represent the angle. In the original formula, the angle is measured in degrees. In response to your suggestion, we have modified the formula to make it easier for readers to understand.

Changes: n=0.5×[((u_α+u_β))/(〖sin〗^(-1)⁡√(p_1 )-〖sin〗^(-1)⁡√(p_2 ) )]^2

Q3: Page 11 Line 190.Note if the equation is to be quoted, definitions of what uα and uβ represent are required.

Response: Accordingly, we have been defined uα，uβ and added description of uα，uβ on page 11 line 189-190.

Changes: uα and uβ correspond to the μ value of Type I and Type II errors , respectively, uα = 1.96; uβ = 0.842.

Q4: Page 12 Line 213 Table 1. Column 1 rows may be better labelled PCN, Test, Control or PCN, Yes, No.

Table 1 As the L and C groups are allocated at random, statistical testing is not required for comparing differences of age, BMI and ASA as, by definition, such differences are random. So, best to omit the P-value row.

Response: Accordingly, we have revised the label name of the test group to “Group T” and deleted the p-value row. 

Q5: Table 1. As this is describing the patients, rather than the SD better to quote the minimum and maximum values of each variable.

Response: After considering this advice, we think that SD can better reflect the distribution of the variable than the minimum and maximum values of each variable, and we have consulted several literatures related to nerve block published by PLoS One, in which SD is also used to describe the variation. However, if you think this item must be modified, please let us know, and we will adopt your opinions to make changes.

References:

[1] Kang Q, Wu L, Liu Y, Zhang X. Ultrasound-guided medial branch of the superior laryngeal nerve block to reduce peri-operative opioids dosage and accelerate patient recovery. PLoS One. 2023; 18 (12): e0295127. doi: 10.1371/journal.pone.0295127

[2] Zhipeng L, Meiyi H, Meirong W, Qunmeng J, Zhenhua J, Yuezhen H, et al. Ultrasound-guided internal branch of superior laryngeal nerve block on postoperative sore throat: A randomized controlled trial. PLoS One. 2020; 15 (11): e0241834. doi: 10.1371/journal.pone.0241834

Q6: Page 12 Table 2. As the focus of the trial is on the difference L − C, this table should give these differences for each of T1 to T4, that is, -1.98, -1.08, -3.38 and -2.77. Then quote the 95% CI for each of these differences plus the P-values already quoted.Usual to quote P-values to 2 significant figures, hence 0.47, 0.68, 0.20 and 0.29. This comment also applies to other places in the text.

Response: Accordingly, we have added the difference T – C and 95% CI for each of the these differences for T1 to T4. Furthermore, all p-values have been presented to two significant figures.

Q7: Page 12-13 Table 3. Same comments as for Table 2.

Response: Accordingly, we have added the difference T – C and 95% CI for each of the these differences for T1 to T4. Furthermore, all p-values have been presented to two significant figures.

Q8: Page 13 Tables 4 & 5. I think it would be more helpful to quote the minimum and maximum scores rather than Q1 and Q3.

Response: After taking this advice into account, we believe that Q1 and Q3 can reflect the distribution of the variable better than the minimum and maximum values of each variable. While deliberating this change, we have consulted several literatures related to nerve block published by PLoS One, in which IQR is also used to describe the variation. If you think this item must be modified, please let us know, and we will make the relevant changes.

References :

[3] Ranganath YS, Ramanujam V, Onodera Y, Qunmeng J, Zhenhua J, Yuezhen H, et al. Impact of paravertebral blocks on analgesic and non-analgesic outcomes after video-assisted thoracoscopic surgery: A propensity matched cohort study. PLoS One. 2021; 16. doi: 10.1371/journal.pone.0252059

Q9: Page 14-15 Tables 6 & 7. More helpful to quote the minimum and maximum scores rather than Q1 and Q3.

Response: After considering this advice, we believe that utilizing the Q1 and Q3 can provide a more accurate reflection of the variable's distribution compared to relying solely on the minimum and maximum values. During our deliberation on this adjustment, we referred to several studies on nerve blocks published by PLoS One, where the IQR was commonly employed for this description. If you still believe that this aspect necessitates modification, please inform us, and we will gladly incorporate your suggestions.

References:

[3] Ranganath YS, Ramanujam V, Onodera Y, Qunmeng J, Zhenhua J, Yuezhen H, et al. Impact of paravertebral blocks on analgesic and non-analgesic outcomes after video-assisted thoracoscopic surgery: A propensity matched cohort study. PLoS One. 2021; 16. doi: 10.1371/journal.pone.0252059

Q10 Page 15 Table 7, row 2. Time taken by patients with high Alderete scores refers to a subset of the 80 patients, in which case the numbers concerned in each group need to be quoted here. Also, the difference in groups needs to be quoted with the 95% CI. Since the distribution of time is very skew, comparisons of medians are required. I do not know if these can be implemented in SPSS Statistics 23.0 which is used by the authors for their analysis.

Response: Alderte score is a scoring system used to evaluate the recovery of patients after anesthesia, and the full score is 10 points. Alderte score ≥ 9 represents complete recovery of patients. Therefore, the Alderte score of all patients will be ≥ 9, and there are no patients with Alderte score <9. We analyzed the time for achieving an Aldrete score of ≥9 of the patients and modified the description of the Aldrete score in Table 7 to facilitate reader understanding. We have added the difference L – C (95% CI) of Alderte score in Table 7.

Q11: Although, there is considerable interest in the differences between randomised groups at T1, T2, T3 and T4 the following overall comparison may be useful to include.

Response: Accordingly, in Table 7, we have provided a general comparison of the operating times, Alderte score≥9 min, operator satisfaction, patient satisfaction, and patient willingness to return.

Reviewer #2:

Q1: This type of treatment procedure has a very short duration, and in the past, we have achieved good results by combining short-term sedatives with low-dose analgesics. However, in this study, the two groups did not routinely use analgesic drugs, so it is reasonable for the control group to have a higher VAS score in T4.

Response: In terms of analgesic drug administration, both groups were administered 0.03 mg/kg of nalbuphine for intravenous analgesia, as stated in the article on Page 8 Line 124.

Q2: Compared to administering low-dose analgesics, ultrasound-guided paracervical nerve block requires additional equipment, operator capabilities, and more potential complications. Is it worth doing this in clinical work? Whether nerve block is better than local anesthesia or the use of low-dose analgesics should be more discussed. If a simpler method can be used to achieve the same effect, nerve block is obviously a more complex and risky operation.

Response: While the ultra-guided parallel nerve block does require additional equipment and places higher demands on the operator, this study revealed notable benefits for patients:

1. Test group patients experienced a significant decrease in VAS score at T3 and T4.

2. At T2, there was a significant reduction in the body movement scores of the test group patients.

3. Both patient and operator satisfaction were high in the test group. 

4. The test group exhibited a greater willingness to return. 

5. Moreover, the parallel nerve block, completed under ultrasound guidance, offers the advantage of constant visualization of the puncture needle throughout the entire process of implementing the nerve block, avoiding complications. Notably, there were no nerve block related complications. 

We have added the relevant information to the Discussion to further highlight the benefits of the method compared with alternative procedures, page 20-21, lines 302-314.

Changes: Research indicates that conscious sedation anesthesia can be used in patients with gynecological tumors undergoing ISBT treatment. However, when the sedation effect disappears and the instrument is removed, patients often exhibit moderate pain, subsequently requiring high doses of analgesics and potential psychosocial treatment [20]. Studies have also investigated a combination of propofol and ketamine for anesthesia. Although it can yield effective anesthesia, the subsequent incidence of nausea and vomiting in patients is high, which significantly increases their discomfort [21]. Moreover, following applicator placement, it necessitates transfer between the operating, imaging, and radiation therapy rooms. Such repeated movement can further increase the patient's experience of pain stimulation. Therefore, adopting the anesthesia method of remifentanil sedation combined with ultrasound-guided paracervical nerve block presents as a more suitable choice.

Q3: Is it common to use BIS monitoring for only 5 minutes of operation? Is it necessary or a waste of resources?

Response: The use of BIS monitoring aimed to maintain uniform anesthesia depth across the two groups, mitigating potential outcome disparities due to different anesthesia depths. In clinical settings, the decision to use BIS monitoring is based on factors such as the patient's age and physical condition.

---

## [Decision Letter · Decision Letter 1]

15 Jul 2024

PONE-D-23-40504R1Pain Score reduction with the use of ultrasound-guided paracervical nerve block in patients with cervical cancer undergoing intracavitary brachytherapy: A randomized controlled trialPLOS ONE

Dear Dr. Zhou,

Thank you for submitting your manuscript to PLOS ONE. After careful consideration, we feel that it has merit but does not fully meet PLOS ONE’s publication criteria as it currently stands. Therefore, we invite you to submit a revised version of the manuscript that addresses the points raised during the review process.

We look forward to receiving your revised manuscript.

Kind regards,

Olga A Sukocheva, PhD

Academic Editor

PLOS ONE

Journal Requirements:

Additional Editor Comments:

Please address reviewers' comments as much as logically possible.

Reviewers' comments:

Reviewer's Responses to Questions

**Comments to the Author**

1. If the authors have adequately addressed your comments raised in a previous round of review and you feel that this manuscript is now acceptable for publication, you may indicate that here to bypass the “Comments to the Author” section, enter your conflict of interest statement in the “Confidential to Editor” section, and submit your "Accept" recommendation.

Reviewer #1: All comments have been addressed

Reviewer #2: All comments have been addressed

Reviewer #3: All comments have been addressed

2. Is the manuscript technically sound, and do the data support the conclusions?

Reviewer #1: Yes

Reviewer #2: Partly

Reviewer #3: Yes

3. Has the statistical analysis been performed appropriately and rigorously? 

Reviewer #1: Yes

Reviewer #2: Yes

Reviewer #3: Yes

4. Have the authors made all data underlying the findings in their manuscript fully available?

Reviewer #1: Yes

Reviewer #2: Yes

Reviewer #3: Yes

5. Is the manuscript presented in an intelligible fashion and written in standard English?

Reviewer #1: Yes

Reviewer #2: No

Reviewer #3: Yes

6. Review Comments to the Author

Reviewer #1: Accept

The authors have taken note of the suggestions made in my previous review. However, there remains one small point which is of little consequence to the interpretation of their results. I suggested that for the descriptive purposes of Table 1 it is more useful for the reader to see the minimum and maximum values of age and BMI. I agree with the authors, that many publications would use SD but I don’t agree that it is best practice.

Reviewer #2: I don't believe the author's revisions meet the standards for journal publication. The benefits of using ultrasound-guided block techniques in this context are not significant when compared to the investments in technology, equipment, risk, and time. The changes in VAS scores are worth further investigation both in terms of statistical and clinical significance

Reviewer #3: 1. Original Submission

Recommendation to the author and editor:

Minor revision

Title: Manuscript ID: PONE-D-23-40504R1 entitled " Pain Score reduction with the use of ultrasound-guided paracervical nerve block in patients with cervical cancer undergoing intracavitary brachytherapy: A randomized controlled trial"

Article Type: Clinical Trial ID: ChiCTR2300071580

2. Comments to the Corresponding Author:

COPE Ethical guidelines followed during the review process,

The manuscript addresses this study that aimed to evaluate the safety and effectiveness of ultrasound-guided paracervical nerve blocks for pain management in cervical cancer patients post-implantation. Authors conducted at the Fourth Hospital of Hebei Medical University from July to October 2023, the single-center randomized controlled trial involved eighty cervical cancer patients undergoing post-implantation treatment. In this clinical trial, authors segregated the patients, who were randomly assigned to two groups: one receiving paracervical nerve blocks (Group T) and the other not receiving them (Group C). Primary measures included visual analog scale (VAS) scores and patient body movement scores at various stages of the procedure. Additionally, willingness to undergo further treatment, operation time, incidence of hypoxemia, occurrence of nausea and vomiting, adverse events related to the circulatory system, patient satisfaction, operator satisfaction, and time to reach an Aldrete score of ≥ 9 were assessed.

Authors described that VAS scores were significantly lower in Group T at T3 and T4 compared to Group C. Similarly, body movement scores were significantly lower in Group T at T2 and T4. Group T also demonstrated higher postoperative satisfaction and a greater willingness to undergo further treatment than Group C. The findings indicate that ultrasound-guided paracervical nerve blocks effectively reduce pain in cervical cancer patients undergoing post-implantation treatment and enhance their inclination to pursue further treatment.

Comments:

Overview and general recommendation:

The paper was well written. I have no more questions as I observed the level of significance among different groups include T1-T4 were satisfactory and yet, proofreading can enhance the quality of the manuscript. Several sentences need rewriting to make the readers comfortable when reading this. Results should be explained vividly for the readers. Good to see that ‘lost to follow up’ in the CONSORT is zero.

1. Authors should expand the discussion part with additional content.

2. In the Figure 2, swap the content in Chinese language with English or Mention a note below the figure. All the figure legends are very simple and authors should expand the legends with significance of the figure to the study.

3. Mention trial number in the abstract : ChiCTR2300071580 [https://clin.larvol.com/trial-detail/ChiCTR2300071580]

3. Conclusion should be explained vividly

4. Mention about the authors future directions depending on the results of this trial: For instance, Future trials could explore several avenues to build on these findings. Firstly, a larger sample size and multi-center trials would provide more robust data and generalizability. Additionally, investigating the long-term effects of paracervical nerve blocks on patient outcomes and quality of life would be valuable.

5. Comparing the efficacy of ultrasound-guided paracervical nerve blocks with other pain management techniques could further refine treatment protocols. Moreover, assessing the cost-effectiveness and potential side effects in a more diverse patient population would be beneficial. Finally, integrating advanced imaging technologies and refining nerve block techniques could enhance precision and effectiveness in pain management for cervical cancer patients.

**Thank you**

7. PLOS authors have the option to publish the peer review history of their article (what does this mean?). If published, this will include your full peer review and any attached files.

Reviewer #1: No

Reviewer #2: No

Reviewer #3: **Yes: **Narasimha M Beeraka, PhD

---

## [Author Response · Author response to Decision Letter 1]

16 Aug 2024

August 10, 2024

Marianne Clemence

Staff Editor

PLoS One

Dear Editors:

We wish to re-submit the manuscript entitled “Pain score reduction with the use of ultrasound-guided paracervical nerve block in patients with cervical cancer undergoing intracavitary brachytherapy: A randomized controlled trial.” The manuscript ID is PONE-D-23-40504. 

We thank you and the reviewers for your thoughtful suggestions and insights. The manuscript has benefited from these insightful suggestions. I look forward to working with you and the reviewers to move this manuscript closer to publication in the PLoS One.

We have studied the comments carefully and have made the necessary corrections in accordance with the reviewers’ suggestions. Revised portions of the text are marked in the manuscript. The point-by-point responses to all comments have been prepared and given below. 

Thank you for your consideration. I look forward to hearing from you.

Sincerely,

Chao Zhou,

Department of Anesthesiology, The Fourth Hospital of Hebei Medical University

zhouchao870607@163.com

+8618531179002

Response:

Q1: Authors should expand the discussion part with additional content.

Response: Thank you for your recommendation. We have supplemented the Discussion section and reorganized its content to make it easily understandable by the readers.

Q2: In the Figure 2, swap the content in Chinese language with English or Mention a note below the figure. All the figure legends are very simple and authors should expand the legends with significance of the figure to the study.

Response: Thank you for your advice. The Chinese content in the image is automatically generated by the ultrasound machine and has no meaning. However, to avoid any misunderstandings among readers, we have removed that from the image. In addition, we have expanded the figure legends for clarity.

 Q3: Mention trial number in the abstract: ChiCTR2300071580 [https://clin.larvol.com/trial-detail/ChiCTR2300071580].

Response: Thank you for your suggestion. We have mentioned the clinical trial identifier in the Abstract.

 Q4: Conclusion should be explained vividly.

Response: Thank you for your advice. We have partly rewritten the Discussion and provided sufficient explanations for the conclusions therein.

Q5: Mention about the authors future directions depending on the results of this trial: For instance, Future trials could explore several avenues to build on these findings. Firstly, a larger sample size and multi-center trials would provide more robust data and generalizability. Additionally, investigating the long-term effects of paracervical nerve blocks on patient outcomes and quality of life would be valuable. Comparing the efficacy of ultrasound-guided paracervical nerve blocks with other pain management techniques could further refine treatment protocols. Moreover, assessing the cost-effectiveness and potential side effects in a more diverse patient population would be beneficial. Finally, integrating advanced imaging technologies and refining nerve block techniques could enhance precision and effectiveness in pain management for cervical cancer patients

Response: Thank you for your recommendations. We have added our outlook on the research at the end of the Discussion section.

---

## [Editor Report · Decision Letter 2]

28 Aug 2024

Pain Scores reduction with the use of ultrasound-guided paracervical nerve block in patients with cervical cancer undergoing intracavitary brachytherapy: A randomized controlled trial

PONE-D-23-40504R2

Dear Dr. Chao Zhou,

We’re pleased to inform you that your manuscript has been judged scientifically suitable for publication and will be formally accepted for publication once it meets all outstanding technical requirements.

Kind regards,

Olga A Sukocheva, PhD

Academic Editor

PLOS ONE

Additional Editor Comments (optional):

Authors addressed all reviewers' comments properly.

The revised version of this manuscript can be accepted and published. Minor English language check is required.
---

## [Editor Report · Acceptance letter]

15 Oct 2024

PONE-D-23-40504R2 

PLOS ONE

Dear Dr. Zhou, 

I'm pleased to inform you that your manuscript has been deemed suitable for publication in PLOS ONE. Congratulations! Your manuscript is now being handed over to our production team.

Kind regards, 

on behalf of

Dr. Olga A Sukocheva 

Academic Editor

PLOS ONE